# Joint inference and input optimization in equilibrium networks

**Swaminathan Gurumurthy**[*]
Carnegie Mellon University

**Shaojie Bai**
Carnegie Mellon University

**J. Zico Kolter**
Carnegie Mellon University
Bosch Center for AI

**Zachary Manchester**
Carnegie Mellon University

## Abstract

Many tasks in deep learning involve optimizing over the *inputs* to a network to minimize or maximize some objective; examples include optimization over latent spaces in a generative model to match a target image, or adversarially perturbing an input to worsen classifier performance. Performing such optimization, however, is traditionally quite costly, as it involves a complete forward and backward pass through the network for each gradient step. In a separate line of work, a recent thread of research has developed the deep equilibrium (DEQ) model, a class of models that foregoes traditional network depth and instead computes the output of a network by finding the fixed point of a single nonlinear layer. In this paper, we show that there is a natural synergy between these two settings. Although, naively using DEQs for these optimization problems is expensive (owing to the time needed to compute a fixed point for each gradient step), we can leverage the fact that gradient-based optimization can *itself* be cast as a fixed point iteration to substantially improve the overall speed. That is, we *simultaneously* both solve for the DEQ fixed point *and* optimize over network inputs, all within a single "augmented" DEQ model that jointly encodes both the original network and the optimization process. Indeed, the procedure is fast enough that it allows us to efficiently *train* DEQ models for tasks traditionally relying on an "inner" optimization loop. We demonstrate this strategy on various tasks such as training generative models while optimizing over latent codes, training models for inverse problems like denoising and inpainting, adversarial training and gradient based meta-learning.

## 1 Introduction

Many settings in deep learning involve optimization over the inputs to a network to minimize some desired loss. For example, for a "generator" network $G : \mathcal{Z} \to \mathcal{X}$ that maps from latent space $\mathcal{Z}$ to an observed space $\mathcal{X}$, it may be desirable to find a latent vector $z \in \mathcal{Z}$ that most closely produces some target output $x \in \mathcal{X}$ by solving the optimization problem (e.g. [10, 13])

$$\underset{z \in \mathcal{Z}}{\text{minimize}} \ \|x - G_\theta(z)\|_2^2. \tag{1}$$

As another example, constructing adversarial examples for classifiers [28, 53] typically involves optimizating over a perturbation to a given input; i.e., given a classifier network $g : \mathcal{X} \to \mathcal{Y}$, task loss

---

[*]Correspondence to: Swaminathan Gurumurthy <sgurumur@andrew.cmu.edu>
Code available at `https://github.com/locuslab/JIIO-DEQ`

35th Conference on Neural Information Processing Systems (NeurIPS 2021).

$\ell : \mathcal{Y} \to \mathbb{R}_+$, and a sample $x \in \mathcal{X}$, we want to solve

$$\underset{\|\delta\| \leq \epsilon}{\text{maximize}} \ \ell(g(x + \delta)). \tag{2}$$

More generally, a wide range of inverse problems [10] and other auxiliary tasks [22, 3] in deep learning can also be formulated in such a manner.

Orthogonal to this line of work, a recent trend has focused on the use of an *implicit layer* within deep networks to avoid traditional depth. For instance, Bai et al. [5] introduced deep equilibrium models (DEQs) which instead treat the network as repeated applications of a single layer and compute the output of the network as a solution to an equilibrium-finding problem instead of simply specifying a sequence of non-linear layer operations. Bai et al. [5] and subsequent work [6] have shown that DEQs can achieve results competitive with traditional deep networks for many realistic tasks.

In this work, we highlight the benefit of using these implicit models in the context of input optimization routines. Specifically, because optimization over inputs itself is typically done via an iterative method (e.g., gradient descent), we can combine this optimization fixed-point iteration *with* the forward DEQ fixed point iteration all within a single "augmented" DEQ model that *simultaneously* performs forward model inference as well as optimization over the inputs. This enables the models to more quickly perform both the inference and optimization procedures, and the resulting speedups further allow us to *train* networks that use such "bi-level" fixed point passes. In addition, we also show a close connection between our proposed approach and the primal-dual methods for constrained optimization.

We illustrate our methods on 4 tasks that span across different domains and problems: 1) training DEQ-based generative models while optimizing over latent codes; 2) training models for inverse problems such as denoising and inpainting; 3) adversarial training of implicit models; and 4) gradient-based meta-learning. We show that in all cases, performing this simultaneous optimization and forward inference accelerates the process over a more naive inner/outer optimization approach. For instance, using the combined approach leads to a 3.5-9x speedup for generative DEQ networks, a 3x speedup in adverarial training of DEQ networks and a 2.5-3x speedup for gradient based meta-learning. In total, we believe this work points to a variety of new potential applications for optimization with implicit models.

## 2   Related Work

**Implicit layers.**   Layers with implicitly defined depth have gained tremendous popularity in recent years[46, 19, 29]. Rather than a static computation graph, these layers define a condition on the output that the model must satisfy, which can represent "infinite" depth, be directly differentiated through via the implicit function theorem [47], and are memory-efficient to train. Some recent examples of implicit layers include optimization layers [16, 1], deep equilibrium models[5, 6, 68, 40, 52], neural ordinary differential equations (ODEs) [14, 18, 61], logical structure learning [67], and continuous generative models [30].

In particular, deep equilibrium models (DEQs) [5] define the output of the model as the fixed point of repeated applications of a layer. They compute this using black-box root-finding methods[5] or accelerated fixed-point iterations [36] (e.g., Broyden's method [11]). In this work, we propose an efficient approach to perform input optimization with the DEQ by *simultaneously* optimizing over the inputs and solving the forward fixed point of an equilibrium model as a joint, augmented system. As related work, Jeon et al. [36] introduce fixed point iteration networks that generalize DEQs to repeated applications of gradient descent over variables. However, they don't address the specific formulation presented in this paper, which has a number of practical use cases (e.g., adversarial training). Lu et al. [52] proposes an implicit version of normalizing flows by formulating a joint root-finding problem that defines an invertible function between the input $x$ and output $z^\star$. Perhaps the most relevant approach to our work is Gilton et al. [26], which specifically formulates inverse imaging problems as a DEQ model. In contrast, our approach focuses on solving input optimization problems where the network of interest is *already* a DEQ, and thus the combined optimization and forward inference task leads to a substantially different set of update equations and tradeoffs.

**Input optimization in deep learning.** Many problems in deep learning can be framed as optimizing over the inputs to minimize some objective . Some canonical examples of this include finding adversarial examples [53, 45], solving inverse problems [10, 13, 56], learning generative models [9, 72], meta-learning [58, 22, 74, 32] etc. For most of these examples, input optimization is typically done using gradient descent on the input, i.e., we feed the input through the network and compute some loss, which we minimize by optimizing over the input with gradient descent. While some of these problems might not require differentiating through the entire optimization process, many do (introduced below), and can further slow down training and impose massive memory requirements.

Input optimization has recently been applied to train generative models. Zadeh et al. [72], Bojanowski et al. [9] proposed to train generator networks by jointly optimizing the parameters and the latent variables corresponding to each example. Similarly, optimizing a latent variable to make the corresponding output match a target image is common in decoder-only models like GANs to get correspondences [10, 39], and has been found useful to stabilize GAN training [71]. However, in all of these cases, the input is optimized for just a few (mostly 1) iterations. In this work, we present a generative model, where we optimize and find the *optimal* latent code for each image at each training step. Additionally, Bora et al. [10], Chang et al. [13] showed that we can take a pretrained generative model and use it as a prior to solve for the likely solutions to inverse problems by optimizing on the input space of the generative model (i.e., unsupervised inverse problem solving). Furthermore, Diamond et al. [15], Gilton et al. [25], Gregor and LeCun [31] have shown that networks can also be trained to solve specific inverse problems by effectively unrolling the optimization procedure and iteratively updating the input. We demonstrate our approach in the unsupervised setting as in Bora et al. [10], Chang et al. [13], but also show flexible extension of our framework to train implicit models for supervised inverse problem solving.

Another crucial application of input optimization is to find adversarial examples [64, 28]. This manifests as optimizing an objective that incentivices an incorrect prediction by the classifier, while constraining the input to be within a bounded region of the original input. Many attempts have been made on the defense side [57, 37, 65, 69]. The most successful strategy thus far has been adversarial training with a projected gradient descent (PGD) adversary [53] which involves training the network on the adversarial examples computed using PGD *online during training*. We show that our joint optimization approach can be easily applied to this setting, allowing us to train implicit models to perform competitively with PGD in guaranteeing adversarial robustness, but at much faster speeds.

While the examples above were illustrated with non-convex networks, attempts have also been made to design networks whose output is a convex function of the input [2]. This allows one to use more sophisticated optimization algorithms, but usually at a heavy cost of model capacity. They have been demonstrated to work in a variety of problems including multi-label prediction, image completion [2], learning stable dynamical systems [44] and optimal transport mappings [54], MPC [12], etc.

## 3 Joint inference and input optimization in DEQs

Here we present our main methodological contribution, which sets up an augmented DEQ that jointly performs inference and input optimization over an existing DEQ model. We first define the base DEQ model, and then illustrate a joint approach that simultaneously finds it's forward fixed point and optimizes over its inputs. We discuss several methodological details and extensions.

### 3.1 Preliminaries: DEQ-based models

To begin with, we recall the deep equilibrium model setting from Bai et al. [5], but with the notation slightly adapted to better align with its usage in this paper. Specifically, we consider an *input-injected* layer $f_\theta : \mathcal{Z} \times \mathcal{X} \to \mathcal{Z}$ where $\mathcal{Z}$ denotes the hidden state of the network, $\mathcal{X}$ denotes the input space, and $\theta$ denotes the parameters of the layer. Given an input $x \in \mathcal{X}$, computing the forward pass in a DEQ model involves finding a fixed point $z^\star(x) \in \mathcal{Z}$, such that

$$z_\theta^\star(x) = f_\theta(z_\theta^\star(x), x), \tag{3}$$

which (under proper stability conditions) corresponds to the "infinite depth" limit of repeatedly applying the $f_\theta$ function. We emphasize that under this setting, we can effectively think of $z_\theta^\star$ *itself*

as the implicitly defined network (which thus is also parameterized by $\theta$), and one can differentiate through this "network" via the implicit function theorem [8, 47].

The fixed point of a DEQ could be computed via the simple forward iteration

$$z^+ := f_\theta(z, x) \tag{4}$$

starting at some artibrary initial value of $z$ (typically 0). However, in practice DEQ models will typically compute this fixed point not simply by iterating the function $f_\theta$, but by using a more accelerated root-finding or fixed-point approach such as Broyden's method [11] or Anderson acceleration [4, 66]. Further, although little can be said about e.g., the existence or uniqueness of these fixed points *in general* (though there do exist restrictive settings where this is possible [68, 59, 23]), in practice a wide suite of techniques have been used to ensure that such fixed points exist, can be found using relatively few function evaluations, and are able to competitively model large-scale tasks [5, 6].

## 3.2 Joint inference and input optimization

Now we consider the setting of performing *input optimization* for such a DEQ model. Specifically, consider the task of attempting to optimize the input $x \in \mathcal{X}$ to minimize some loss $\ell : \mathcal{Z} \times \mathcal{Y} \to \mathbb{R}_+$.

$$\underset{x \in \mathcal{X}}{\text{minimize}} \ \ell(z_\theta^\star(x), y) \tag{5}$$

where $y \in \mathcal{Y}$ represents the data point. To solve this, we typically perform such an optimization via e.g., gradient descent, which repeats the update

$$x^+ := x - \alpha \left( \frac{\partial \ell(z_\theta^\star(x), y)}{\partial x} \right)^\top \tag{6}$$

until convergence, where we use term $z^\star$ alone to denote the fixed output of the network $z_\theta^\star$ (i.e., just as a fixed output rather than a function). Using the chain rule and the implicit function theorem, we can further expand update (6) using the following analytical expression of the gradient:

$$\frac{\partial \ell(z_\theta^\star(x), y)}{\partial x} = \frac{\partial \ell(z^\star, y)}{\partial z^\star} \frac{\partial z_\theta^\star(x)}{\partial x} = \frac{\partial \ell(z^\star, y)}{\partial z^\star} \left( I - \frac{\partial f_\theta(z^\star, x)}{z^\star} \right)^{-\top} \frac{\partial f_\theta(z^\star, x)}{\partial x} \tag{7}$$

Thinking about $z_\theta^\star$ as an implicit function of $x$ permits us to combine the fixed-point equation in Eq. 4 (on $z$) with this input optimization update (on $x$), thus performing a joint forward update:

$$\begin{bmatrix} z^+ \\ x^+ \end{bmatrix} := \begin{bmatrix} f_\theta(z, x) \\ x - \alpha \left( \frac{\partial f_\theta(z,x)}{\partial x} \right)^\top \left( I - \frac{\partial f_\theta(z,x)}{\partial z} \right)^{-\top} \left( \frac{\partial \ell(z,y)}{\partial z} \right)^\top \end{bmatrix} \tag{8}$$

It should be apparent that, if both iterates converge, then they have converged to a simultaneous fixed point $z^\star$ and an optimal $x^\star$ value for the optimization problem (5). However, simply performing this update can still be inefficient, because computing the inverse Jacobian in Eq. (7) is expensive and typically computed via an iterative update – namely, we would first compute the variable $\mu = \left( I - \frac{\partial f_\theta(z,x)}{\partial z} \right)^{-\top} \left( \frac{\partial \ell(z,y)}{\partial z} \right)^\top$ via the following iteration (i.e., a Richardson iteration [60]):

$$\mu^+ := \left( \frac{\partial f_\theta(z, x)}{\partial z} \right)^\top \mu + \left( \frac{\partial \ell(z, y)}{\partial z} \right)^\top. \tag{9}$$

Therefore, to efficiently solve the joint inference and input optimization problem, we propose combining *all three* iterative procedures into the update

$$\begin{bmatrix} z^+ \\ \mu^+ \\ x^+ \end{bmatrix} := \begin{bmatrix} f(z, x) \\ \left( \frac{\partial f_\theta(z,x)}{\partial z} \right)^\top \mu + \left( \frac{\partial \ell(z,y)}{\partial z} \right)^\top \\ x - \alpha \left( \frac{\partial f_\theta(z,x)}{\partial x} \right)^\top \mu \end{bmatrix} \tag{10}$$

Like Eq. (8), if this joint process converges to a fixed point, then it corresponds to a simultaneous optimum of both the inference and optimization processes. Such a formulation is especially appealing, as the iteration (10) is *itself* just an *augmented DEQ network* $v_\theta^\star(y)$ (i.e., with input injection $y$) whose forward pass optimizes on a joint inference-optimization space $v = (x, \mu, z)$. Moreover, we can use standard techniques to differentiate through *this* process, though there are also optimizations

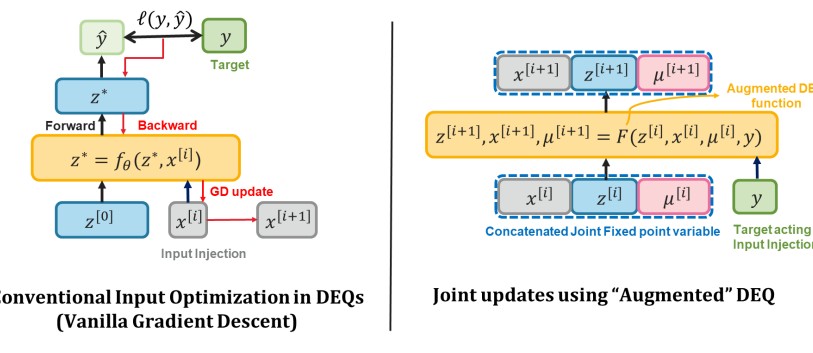

**Conventional Input Optimization in DEQs**
**(Vanilla Gradient Descent)**

**Joint updates using "Augmented" DEQ**

.

Figure 1: Left: Performing each gradient update on DEQ inputs requires a fixed point computation in the forward and backward pass. Right: Solving the 3 fixed points simultaneously as an "augmented" DEQ where the targets $y$ act as input and the function $F$ represents the joint fixed point updates in Eq. 11

we can apply in several settings that we discuss below. This is in contrast to prior works where $f_\theta$ is an explicit deep neural network, where the model forward-pass and optimization processes are disentangled and have to be dealt with separately. We illustrated this in Figure 1 where the figure on the left shows the input optimization naively performed using gradient descent in DEQs v/s the figure on the right which shows the joint updates performed using the augmented DEQ network.

As a final note, we mention that, in practice, just as the gradient-descent update has a step size $\alpha$, it is often beneficial to add similar "damping" step sizes to the other updates as well. This leads to the full iteration over the augmented DEQ

$$\begin{bmatrix} z^+ \\ \mu^+ \\ x^+ \end{bmatrix} := \begin{bmatrix} (1-\alpha_z)z + \alpha_z f(z,x) \\ (1-\alpha_\mu)\mu + \alpha_\mu \left( \left(\frac{\partial f_\theta(z,x)}{\partial z}\right)^\top \mu + \left(\frac{\partial \ell(z,y)}{\partial z}\right)^\top \right) \\ x - \alpha_x \left(\frac{\partial f_\theta(z,x)}{\partial x}\right)^\top \mu \end{bmatrix} \tag{11}$$

Finally, in order to speed up convergence, as is common in DEQ models, we apply a more involved fixed point solver, such as Anderson acceleration, on top of this naive iteration. We analyze the effect of these different root-finding approaches in the Appendix.

**Notes on Convergence** Our treatment of the above system as an augmented DEQ allows us to borrow results from [68][7] to ensure convergence of the fixed point iteration. Specifically, if we assume the joint Jacobian of the fixed point iterations we describe are strongly monotone with smoothness parameter $m$ and Lipschitz constant $L$, then by standard arguments (see e.g., Section 5.1 of [62]), the fixed point iteration with step size $\alpha < m/L^2$ will converge. Note that these are substantially weaker rates and constants than required for typical gradient descent or the minimization of locally convex function because the coupling between the three fixed point iterations introduce cross-terms in the joint Jacobian.

Due to these cross-terms, going from the strong monotonicity assumption on the joint fixed point iterations to specific assumptions on $f_\theta$ and $\ell$ is less straightforward. However, empirically, we observed that as long as the step sizes $\alpha$'s were kept reasonably small and the functions $f_\theta$ and $\ell$ were designed appropriately (e.g $\ell$ respecting notions of local convexity and $f_\theta$ with Jacobian eigenvalues less than 1, etc.) the fixed point iterations converged reliably.

### 3.3 Iterpretation as a primal-dual optimization

While the problem above was introduced as an input-optimization problem, its formulation as a joint optimization problem in the augmented DEQ system (10) can also be viewed as a constrained

optimization problem where the DEQ fixed-point conditions are treated as constraints,

$$\underset{x,z}{\text{minimize}} \ \ell(z, y) \ \text{subject to} \ z = f_\theta(z, x) \tag{12}$$

which yield the Lagrangian

$$\underset{x,z}{\text{minimize}} \, \underset{\mu}{\text{maximize}} \ \mathcal{L}(x, z, \mu) \equiv \ell(z, y) + \mu^\top (f_\theta(z, x) - z) \tag{13}$$

and the corresponding KKT conditions

$$\begin{bmatrix} f_\theta(z, x) - z \\ \frac{\partial \mathcal{L}(x,z,\mu)}{\partial z} \\ \frac{\partial \mathcal{L}(x,z,\mu)}{\partial x} \end{bmatrix} = \begin{bmatrix} f_\theta(z, x) - z \\ \frac{\partial \ell(z,y)}{\partial z} + \mu^\top \left( \frac{\partial f_\theta(z,x)}{\partial z} - I \right) \\ \mu^\top \frac{\partial f_\theta(z,x)}{\partial x} \end{bmatrix} = \begin{bmatrix} 0 \\ 0 \\ 0 \end{bmatrix} \in \mathbb{R}^{2n+d} \tag{14}$$

where $\mu$ are the dual variables corresponding to the equality constraints. Rearranging the terms in the KKT conditions of the above problem, introducing the step size parameters $\alpha's$ and treating it as fixed point iteration gives us the updates in Eq. (11). Indeed, performing such iterations is a variation of the classical primal-dual gradient method for solving equality-constrained optimization problems [20, 34, 17].

## 3.4 Outer Optimization (Backward Pass)

A notable advantage of formulating the entire joint inference and input optimization problem as an augmented DEQ $v_\theta^\star(y)$ is that it allows us to abstract away the detailed function of $v$, and simply train parameters $\theta$ of this joint process as an outer optimization problem:

$$\underset{\theta}{\text{minimize}} \ \ell^{\text{outer}}(v_\theta^\star(y), y) \tag{15}$$

where $\ell^{\text{outer}} : \mathcal{X} \times \mathcal{Z} \times \mathcal{Z} \times \mathcal{Y} \to \mathbb{R}_+$

Given the solutions $x^\star, z^\star$, to the inner problem, computing updates to $\theta$ in the outer optimization problem is equivalent to the backward pass of the augmented DEQ and correspondingly is optimized using standard stochastic gradient optimizers like Adam [42]. Thus, as with any other DEQ model, we assume the inner problem was solved to a fixed point, and apply the implicit function theorem to compute the gradients w.r.t the augmented system (11). This gives us a constant memory backward pass which is invariant to the underlying optimizer used to solve the inner problem.

$$\frac{\partial \ell^{\text{outer}}(v_\theta^\star, y)}{\partial \theta} = \frac{\partial \ell^{\text{outer}}(y, v^\star)}{\partial v^\star} \frac{\partial v_\theta^\star}{\partial \theta} = -\frac{\partial \ell^{\text{outer}}(y, v^\star)}{\partial v^\star} \left( \frac{\partial K_\theta(v^\star)}{\partial v^\star} \right)^{-1} \frac{\partial K_\theta(v^\star)}{\partial \theta} \tag{16}$$

where $v = [x, z, \mu]^\top$ and $K_\theta(v) = 0$ represents the KKT conditions from (14). As with the original DEQ, instead of computing the above expression explicitly, we first solve the following linear system to compute $u$ and then substitute it back in the equation above to obtain the full gradient,

$$u^\top = -\frac{\partial \ell^{\text{outer}}(y, v^\star)}{\partial v^\star} \left( \frac{\partial K_\theta(v)}{\partial v} \right)^{-1} \quad \Longleftrightarrow \quad \frac{\partial \ell^{\text{outer}}(y, v^\star)}{\partial v^\star}^\top + \left( \frac{\partial K_\theta(v)}{\partial v} \right)^\top u = 0 \tag{17}$$

Although we can train any joint DEQ in this manner, doing so in practice (e.g., via automatic differentiation), will require double backpropagation, because the definition of $K_\theta(v)$ above already includes vector-Jacobian products, and this expression will require differentiating again. However, in the case that $\ell^{\text{outer}}$ is the *same* as $\ell$ (or in fact where it is the negation of $\ell$), then there exists a substantial simplification of the outer optimization gradient. These cases are indeed quite common, as we may want e.g., to train the parameters of a generative model to minimize the same reconstruction error that we attempt to optimize via the latent variable; or in the case of adversarial examples, the inner adversarial optimization is precisely the negation of the outer objective.

In these cases, we have that

$$\ell^{\text{outer}}(y, v_\theta^\star(y)) = \ell(y, z_\theta^\star(y)) \tag{18}$$

so we have that

$$\frac{\partial \ell^{\text{outer}}(y, v_\theta^\star(y))}{\partial \theta} = \frac{\partial \ell(y, z^\star)}{\partial z^\star} \left( I - \frac{\partial f_\theta(z^\star, x^\star)}{\partial z^\star} \right)^{-1} \frac{\partial f_\theta(z^\star, x^\star)}{\partial \theta} = (\mu^\star)^\top \frac{\partial f_\theta(z^\star, x^\star)}{\partial \theta} \tag{19}$$

In other words, we can compute the exact needed derivatives with respect to $\theta$ by simply *re-using* the converged solution $v^\star$, without the need to double backpropagate through the KKT system. The same considerations, but just negative, apply to the case where $\ell^{\text{outer}} = -\ell$.

# 4 Experiments

As our approach provides a generic framework for joint modeling of an implicit network's forward dynamics and the "inner" optimization over the input space, we demonstrate its effectiveness and generalizability on 4 different types of problems that are popular areas of research in machine learning: generative modeling [43, 27], inverse problems [10, 25, 31], adversarial training [69, 53] and gradient based meta-learning[22, 58] (results for the latter are in the appendix). In all cases, we show that our joint inference and input optimization (JIIO) provides significant speedups over projected gradient descent applied to DEQ models and that the models trained using JIIO achieve results competitive with standard baselines. In all of our experiments, the design of our model layer $f_\theta$ follows from the prior work on multiscale deep equilibrium (MDEQ) models [6] that have been applied on large-scale computer vision tasks, and where we replace all occurrences of batch normalization [35] with group normalization [70] in order to ensure the inner optimization can be done independently for each instance in the batch. We elaborate on the details of the choice of other hyperparameters and design decisions of our model (such as the damping parameters $\alpha$ in the update step (11)), as well as that of the datasets for each task in the Appendix.

We introduce below each problem instantiation, how they fit into our methodology described in Sec. 3.2, and the result of applying the JIIO framework compared to the alternative methods trained in similar settings. Overall, our results provide strong evidence of benefits of performing joint optimizations on implicit models, thus opening new opportunities for future research in this direction.

## 4.1 Generative Modeling

We study the application of JIIO to learning decoder-only generative models that compute the latent representations by directly minimizing the reconstruction loss [72, 9]; i.e., given a decoder network $D$, the latent representation $x$ of a sample $y$ (e.g., an image) is $x = \min_{x \in \mathcal{X}} \|D(x) - y\|_2^2$.[2] Moreover, instead of placing explicit regularizations on the latent space $\mathcal{X}$ (as in VAEs), we follow [24] to directly train the decoder for reconstruction (and then after training, we fit the resulting latents using a simple density model, post-hoc, for sampling). Formally, given sample data $y_1, \ldots, y_n$ (e.g., $n$ images), the generative model we study takes the following form:

$$
\begin{aligned}
&\underset{\theta}{\text{minimize}} \quad \sum_{i=1}^{n} \|y_i - h_\theta(z_i^\star)\|^2 \\
&\text{subject to} \quad x_i^\star, z_i^\star = \underset{x, z:z=f_\theta(z,x)}{\text{argmin}} \quad \|y_i - h_\theta(z)\|^2, \ i = 1, \ldots, n
\end{aligned}
\tag{20}
$$

where $h_\theta$ is a final output layer that transform the activations $z^\star$ to the target dimensionality. We train the MDEQ-based $f_\theta$ with the JIIO framework on standard 64×64 cropped images from CelebA dataset, which consists of 202,599 images. We use the standard train-val-test split as used in Liu et al. [51] and train the model for 50k training steps. We use Fréchet inception distance (FID) [33] to measure the quality of the sampling and test-time reconstruction of the implicit model trained with JIIO and compare with the other standard baselines such as VAEs [43]. The results are shown in Table 1. JIIO-MDEQ refers to the MDEQ model trained using our setup with 40 JIIO iterations in the inner loop during training (and tested with 100 iterations). MDEQ-VAE refers to an equivalent MDEQ model but with an encoder and a decoder trained as a VAE. We observe that our model's generation quality is competitive with, or better than, each of these encoder-decoder based approaches. Moreover, with the joint optimization proposed, JIIO-MDEQ achieves the best reconstruction quality.

We additionally apply JIIO on pre-trained MDEQ-VAEs (i.e., train an MDEQ-based VAE as usual on optimizing ELBO [43], and take the decoder out) for test-time image reconstruction. The result (shown in Table 1) suggests that the reconstructions obtained as a result are better even than the original MDEQ-VAE. In other words, JIIO can be used with general implicit-mode-based decoders at test time even if the decoder wasn't trained with JIIO.

---

[2]This notation differs from the "standard" notation of latent variable models (where the latent variable is typically denoted by $z$). However, because $x, y, z$ all have standard meanings in setting above, we change from the common notation here to be more consistent with the remainder of this paper.

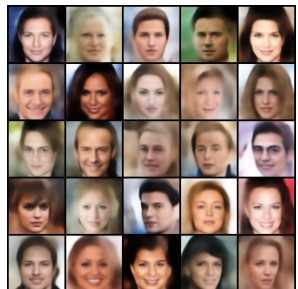

Figure 2: Samples generated with JIIO on a small MDEQ network.

| Model | Generation | Reconstruction |
|---|---|---|
| VAE [43] | 48.12 | 39.12 |
| RAE [24] | **40.96** | 36.01 |
| MDEQ-VAE | 57.15 | 45.81 |
| MDEQ-VAE (w/ JIIO) | - | 42.36 |
| JIIO-MDEQ | 46.82 | **32.52** |

Table 1: Comparison of FID scores attained by standard generative models with our method, which performs joint optimization. We use 40 solver iterations (for the augmented DEQ) to train the JIIO model reported in this table.

One of the key advantages presented by JIIO is the relative speed of optimization over simply running gradient descent (or its adaptive variants like Adam [41]). Table 2 shows our timing results for one optimization run on a single example for various models (averaged over 200 examples). We observe that performing 40 iterations of projected Adam takes more than $9\times$ the time taken by 40 iterations of JIIO, which we used during training and more than $3.5\times$ the time taken by 100 iterations of JIIO which we use for reconstructions at test time (e.g., to produce the results in Table 1, though both of them lead to similar levels of reconstruction loss). Fig 3 shows the reconstruction loss as it evolves across a single optimization run for an MDEQ model trained with JIIO. This again clearly shows that JIIO converges vastly faster (in terms of wall-clock time) than if we handle the inner optimization separately as in prior works, demonstrating the advantage of joint optimization. However, it's interesting to note that JIIO optimization seems somewhat unstable (see Fig. 3) and fluctuates more as well. This seems to be an artifact of the specific acceleration scheme we use (see more details in Appendix A.2).

## 4.2 Inverse Problems

We also extend the setup mentioned in section 4.1 directly to inverse problems. These problems, specifically, can be approached as either an unsupervised or a supervised learning problem, which we discuss separately in this section. To demonstrate how JIIO can be applied, we will be using image inpainting and image denoising as example inverse problem tasks, which was extensively studied in prior works like Chang et al. [13], Gilton et al. [26]. For the inpainting task, we randomly mask a 20x20 window from the image and train the model to adequately fill the missing pixels based on the surrounding image context. For the image denoising tasks, we add random gaussian noise $\varepsilon \sim \mathcal{N}(0, \sigma^2 I)$ with $\sigma = 0.2$ and $\sigma = 0.4$, respectively, to all pixels in the image, and train the model to recover the original image. We use the same datasets and train-test setups as in the generative modeling experiments in Sec. 4.1.

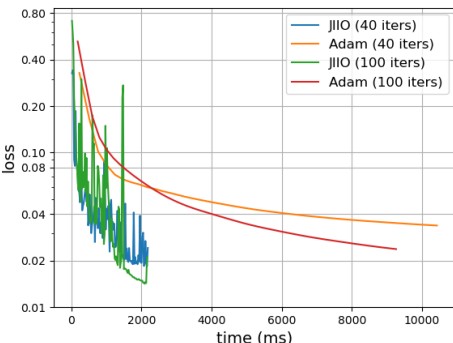

Figure 3: Cost changing with time for Adam v/s JIIO optimization. Tested on models trained with 40 and 100 JIIO iterations respectively

| Model | time taken (ms) |
|---|---|
| PGD : 20 iters | 4360 |
| JIIO : 80 iters | 1401 |

Table 2: Time taken to compute adversarial example of a MDEQ model on MNIST

| Model | time taken (ms) |
|---|---|
| Adam : 40 iters | 7659 |
| JIIO : 40 iters | 862 |
| JIIO : 100 iters | 2156 |

Table 3: Time taken to perform JIIO optimization v/s Adam in the generative modeling/inverse problem experiments

| Task | Model | Inpainting | Denoising ($\sigma = 0.2$) | Denoising ($\sigma = 0.4$) |
|---|---|---|---|---|
| Supervised | AE | 17.9 | 18.72 | 18.32 |
| | MDEQ-AE | 17.06 | 18.58 | 18.49 |
| | JIIO-MDEQ-100 | 16.90 | 18.22 | 17.89 |
| Unsupervised | VAE (Adam) [10] | 15.34 | 15.31 | 15.24 |
| | MDEQ-VAE (Adam) | 16.62 | 16.96 | 16.87 |
| | JIIO-MDEQ-40 | 15.88 | 17.08 | 16.03 |
| | JIIO-MDEQ-100 | 15.87 | 17.86 | 17.55 |

Table 4: Comparison of Median PSNR values for supervised and unsupervised inverse problem solving approaches. The top 3 rows show models that are trained for the specific inverse problem and the latter 5 show pre-trained generative models re-purposed for solving inverse problems

### 4.2.1 Unsupervised inverse problem solving

Bora et al. [10], Chang et al. [13] have showed that we can solve most inverse problems by taking a pre-trained generative model and using that as a prior to solve for the likely solutions to the inverse problems by optimizing on the input space of the generative model. Specifically, given a "generator" network $G : \mathcal{X} \to \mathcal{Y}$, mapping from the latent space $\mathcal{X}$ to an observed space $\mathcal{Y}$, that models the data generating distribution, they show that one can solve any inverse problem by optimizing the following objective:

$$\underset{x \in \mathcal{X}}{\text{minimize}} \ \|\hat{y} - AG(x)\|_2^2. \tag{21}$$

where $\hat{y} = Ay \in \mathcal{Y}$ represents the corrupted data point, $y \in \mathcal{Y}$ is the uncorrupted data and $A : \mathcal{Y} \to \mathcal{Y}$ denotes the measurement matrix that defines the specific type of inverse problem that we try to solve (e.g., for image inpainting, it would be a mask with the missing regions filled in with zeros. For deblurring, it would be a convolution with a gaussian blur operator etc.). They call it unsupervised inverse problem solving. Likewise, we can use the pre-trained generator from section 4.1 to solve most inverse problems by simply solving a slightly modified version of the inner problem in (29):

$$\underset{x,z:z=f_\theta(z,x)}{\text{minimize}} \ \|\hat{y} - Ah_\theta(z)\|^2 \tag{22}$$

In table 4, the unsupervised results for VAE and MDEQ-VAE generators are obtained by optimizing (21) using Adam for 40 iterations, while for the JIIO trained models, we optimize (22) with 100 JIIO iterations. JIIO-MDEQ-40 and JIIO-MDEQ-100 refer to JIIO-MDEQ models trained with 40 and 100 inner-loop iterations respectively. The results in Table 4 show that on all 3 problems, JIIO trained generators produce results comparable to the VAE and MDEQ-VAE generators. Moreover, as shown in section 4.1, JIIO also converges much faster than Adam applied to a MDEQ-VAE generator.

### 4.2.2 Supervised Inverse problem solving

While the unsupervised inverse problem solving works reasonably well, we can also learn models to solve specific inverse problems to obtain better performance. Specifically, given uncorrupted data $y_1, \ldots, y_n$, and the measurement matrix $A$, we can train a network $G_\theta : \mathcal{Y} \to \mathcal{Y}$ mapping from the corrupted sample $\hat{y}_i = Ay_i$ to the uncorrupted sample $y_i$ by minimizing:

$$\underset{\theta}{\text{minimize}} \ \sum_{i=1}^{n} \|y_i - G_\theta(Ay_i)\|^2 \tag{23}$$

Now, instead of modeling $G_\theta$ as an explicit network, we could also model it as a solution to the inverse problem in (22) and the resulting parameters can be trained as follows:

$$\underset{\theta}{\text{minimize}} \ \sum_{i=1}^{n} \|y_i - h(z_i^\star)\|^2$$
$$\text{subject to} \ x_i^\star, z_i^\star = \underset{x,z:z=f_\theta(z,x)}{\text{argmin}} \ \|Ay_i - Ah(z)\|^2, \ i = 1, \ldots, n \tag{24}$$

As shown in Table 4 this yields models competitive with their autoencoder based counterparts, while being better than all the unsupervised approaches. Each of the baseline models in the supervised section of Table 4 are trained by simply optimizing (23) with the corresponding model replacing $G_\theta$. However, note that given the models here are trained on specific inverse problems, one would have to train a new model for each new problem as opposed to the unsupervised approach.

| Datasets | Train (↓) Test (→) | Clean | PGD | JIIO |
|---|---|---|---|---|
| | Clean | $99.45 \pm 0.03$ | $80.1 \pm 1.87$ | $65.88 \pm 4.72$ |
| MNIST | PGD | $99.18 \pm 0.03$ | $96.53 \pm 0.05$ | $95.74 \pm 0.04$ |
| | JIIO | $99.32 \pm 0.09$ | $95.74 \pm 0.22$ | $96.63 \pm 0.58$ |
| | Clean | $78.47 \pm 0.94$ | $2.38 \pm 0.41$ | $3.71 \pm 4.01$ |
| CIFAR | PGD | $54.91 \pm 1.01$ | $37.4 \pm 0.26$ | $36.17 \pm 0.55$ |
| | JIIO | $55.54 \pm 0.82$ | $37.31 \pm 0.67$ | $37.77 \pm 0.84$ |

Table 5: Comparison of adversarial training approaches on L2 norm perturbations with $\epsilon = 1$. The rows represent the training procedure and the columns represent the testing procedure

### 4.3 Adversarial Training

Although the previous two tasks were based on image generation, we note that our approach can be used more generally for input optimization in DEQs and illustrate that by applying it to $\ell_2$ adversarial training on DEQ-based classification models. Specifically, given inputs $x_i, y_i$; $i = 1, \ldots, n$, adversarial training seeks to optimize the objective

$$\underset{\theta}{\text{minimize}} \quad \sum_{i=1}^{n} \max_{\|\delta\|_2 \leq \epsilon} \ell(h_\theta(x_i + \delta_i), y_i) \tag{25}$$

We apply our setting to a DEQ-based classifier $h_\theta(x) = h_\theta(z_\theta^\star(x))$ where $z^\star = f_\theta(z^\star, x)$. In this setting, we embed the iterative optimization over $\delta$ (with projected gradient descent) into the augmented DEQ, and write the problem as

$$\underset{\theta}{\text{minimize}} \quad \sum_{i=1}^{n} \ell(h_\theta(z_i^\star), y)$$
$$\text{subject to } z_i^\star, \delta_i^\star = \underset{z, \delta: z = f_\theta(z, x+\delta), \|\delta\|_2 \leq \epsilon}{\text{argmin}} -\ell(h_\theta(z), y), \ i = 1, \ldots, n \tag{26}$$

Specifically we train MDEQ models on CIFAR10 [48] and MNIST [50] using adversarial training against L2 attacks with $\epsilon = 1$ for 20 epochs and 10 epochs respectively using the standard train-val-test splits. Table 5 shows the robust and clean accuracy of models trained using PGD adversarial training, JIIO adversarial training and vanilla training. We find that models trained using JIIO have comparable robust and clean accuracy on both datasets. Furthermore, when tested on models trained without adversarial training, we observe that JIIO serves as a comparable attack method to PGD. Table 4.2 shows the time taken to find the adversarial example for a single image of MNIST using 20 iterations of PGD and 80 iterations of JIIO. We again observe more than 3x speedups when using JIIO over using PGD while obtaining competitive robust accuracy. However, note that, unlike previous experiments in generative modeling/inverse problems, performing adversarial training with truncated JIIO iterations would lead to significant reduction in robust performance due to the adversarial setting.

## 5 Concluding remarks

We present a novel optimization procedure for jointly optimizing over the input and the forward fixed point in DEQ models and show that, for the same class of models, it is $3 - 9\times$ faster than performing vanilla gradient descent or Adam on the inputs. We also apply this approach to 3 different settings to show it's effectiveness: training generative models, solving inverse problems, adversarial training of DEQs and gradient based meta-learning. In the process, we also introduce an entirely new type of decoder only generative model that performs competitively with it's autoencoder based counterparts.

Despite these features, we note that there is substantial room for future work in these directions. Notably, despite the fact that the augmented joint inference and input optimization DEQ can embed both processes in a "single" DEQ model, in practice these joint models take substantially more iterations to converge as well (often in the range of 50-100) than traditional DEQs (often in 10-20 iterations), and correspondingly often use larger memory caches within methods like Anderson acceleration. Thus, while we make the statement that these augmented DEQs are "just" another DEQ, this relative difficulty in finding a fixed point likely adds challenges to training the underlying models. Thus, despite ongoing work in improving the inference time of typical DEQ models, there is substantial room for improvement here in making these joint models truly efficient.

# 6 Acknowledgements

Swaminathan Gurumurthy and Shaojie Bai are supported by a grant from the Bosch Center for Artificial Intelligence. We further thank Zhichun Huang and Taylor Howell for their help with some implementations and for various brainstorming sessions and feedback throughout the course of this project. We would also like to thank Vladlen Koltun for brainstorming ideas especially in the initial stages of the project.

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
