# A  Additional Discussions on Optimization Stability and Speed

## A.1  Regularization

As was observed in previous work [7, 68], the stability of convergence of a DEQ is directly related to the conditioning of the Jacobian matrix at the equilibrium point. To that end, we primarily adopt the regularization proposed by [7] which upper bounds the implicit model's stability by estimating their trace with the Hutchinson estimator: $\text{tr}(J_z) = \text{tr}\left(\frac{\partial f_\theta(z,x)}{\partial z}\right) = \mathbb{E}_{\epsilon \in \mathcal{N}(0,I)}[\epsilon^\top J_z^\top J_z \epsilon]$. However, our exact implementation is subtly different from the original proposal in that we regularize the Jacobian matrix at a *randomly chosen* iterate along the optimization trajectory $(x^{(k)}, z^{(k)})$ instead of just the last iterate $(x^*, z^*)$. This modification is especially important as the optimization trajectories become long (which is the case for our problems; e.g., which could take >80 iterations), which essentially encourages the Jacobian to be not only stable at the end but also during the root-solving process. Specifically, with this modification, the outer objective of JIIO becomes :

$$\underset{\theta}{\text{minimize}}\ \ell^{\text{outer}}(v_\theta^\star(y), y) + \lambda \mathbb{E}_{\epsilon \in \mathcal{N}(0,1)}[\epsilon^\top J_z^\top J_z \epsilon] \tag{27}$$

where $\lambda$ is the regularization coefficient, and in practice we sample 1 or 2 $\epsilon$'s to produce a Monte-Carlo estimation of the expectation term.

## A.2  Choice of Acceleration method

We perform ablation experiments on the choice of the acceleration methods for performing the joint optimization. Figure 4 shows various approaches that can be used to accelerate JIIO applied to a reconstruction task on a pre-trained MDEQ-VAE decoder. Broyden's method treats its solution as a solution to a root finding problem on the KKT conditions instead of as a minimization problem and hence, given the non-convex nature of the problem, could end up chasing arbitrary stationary points. The Anderson based approaches treat the problem as a minimization problem and hence are able to perform much better. Specifically, Type-I Anderson mixing is usually more unstable (a phenomenon that had been discussed in prior work like Fang and Saad [21] and Zhang et al. [73]), and yet manages to attain the lowest loss values in 100 JIIO iterations. Type-II Anderson, on the other hand, allows for much smoother optimization process although it plateaus at higher loss values. However, since speed was an important point of consideration for our experiments, we nevertheless went with Anderson Type-I over Type-II and picked the output of the optimizer using a heuristic that traded off between a small KKT residual and a small cost. Overall, we observed that the criterion for this iterate selection can be somewhat flexible but preferably kept consistent between training and testing runs when we perform JIIO.

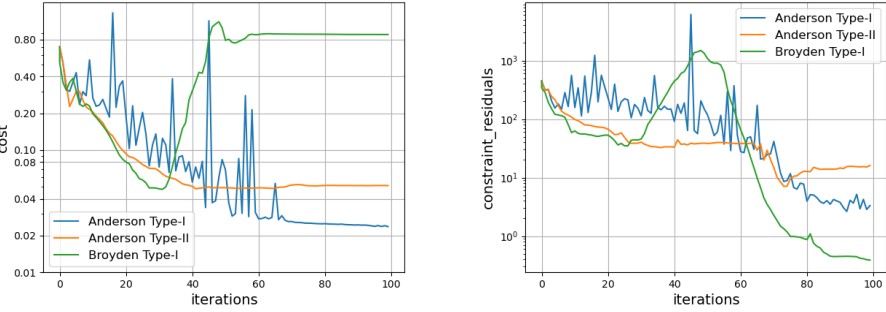

Figure 4: Comparing different acceleration techniques on a MDEQ-VAE decoder

# B Supplementary Experiments and Results

## B.1 Gradient based Meta-Learning

Gradient (or optimization) based meta-learning defines a bi-level optimization problem where the outer loop learns meta-parameters for a distribution of tasks while the inner loop learns task-specific parameters, typically using a small amount of data. This bi-level structure blends itself naturally into the types of input optimization problems we have been looking at. Specifically, taking few-shot learning as the use case, we are given a collection of tasks $\{\mathcal{T}_i\}_{i=1}^N$, each associated with a dataset $\mathcal{D}_i$, from which we can sample two disjoint sets: $\mathcal{D}_i^{tr} = \{(s_{i,k}^{tr}, y_{i,k}^{tr})\}_{k=1}^K$ and $\mathcal{D}_i^{te}\{(s_{i,k}^{te}, y_{i,k}^{te})\}_{k=1}^K$, where each $(s, y)$ is a data-label pair. Gradient based meta-learning for few shot learning problem can be framed as a bi-level optimization problem with input optimization as the inner loop:

$$\begin{aligned} \underset{\theta}{\text{minimize}} \; & \ell(x_i^\star, h_\theta(x_i^\star, s_{i,k}^{te}), y_{i,k}^{te}), \quad i = 1, \ldots, N \\ \text{subject to} \; & x_i^\star = \underset{x_i}{\arg\min} \; \ell(x_i, h_\theta(x_i, s_{i,k}^{tr}), y_{i,k}^{tr}), \quad k = 1, \ldots, K \end{aligned} \tag{28}$$

where $x_i^\star$ are the task specific parameters inferred during the inner-loop optimization and $\theta$ are the meta-parameters. Note that the task specific parameters can be treated as inputs, and thus the inner problem becomes an input optimization problem. In this case, with a DEQ network, the above problem can be modified slightly as:

$$\begin{aligned} \underset{\theta}{\text{minimize}} \; & \ell(x_i^\star, h_\theta(z_{i,k}^{\star (te)}), y_{i,k}^{te}), \quad i = 1, \ldots, N \\ \text{subject to} \; & x_i^\star, z_{i,k}^{\star (tr)}, z_{i,k}^{\star (te)} = \underset{x_i, z_{i,k}^{(tr/te)} = f_\theta(z_{i,k}^{(tr/te)}, x_i)}{\arg\min} \; \ell(x_i, h_\theta(z_{i,k}^{tr}), y_{i,k}^{tr}), \quad k = 1, \ldots, K \end{aligned} \tag{29}$$

Clearly, this modified problem can now to solved using JIIO in the inner loop. Table 6 shows the accuracies obtained by a JIIO-trained DEQ model and various baseline meta-learning approaches like Implicit-MAML [58], MAML [22] and Reptile[55] on the 5-way, 1-shot task on Omniglot [49]. We observe that the JIIO-trained DEQ model achieves comparable accuracy to the baselines. The partially lower accuracy numbers of the DEQ model may be attributed to the fact that we do not differentiate through the fixed point variable $x_i^\star$ while optimizing the outer loop objective. We observed very poor conditioning in our experiments when trying to differentiate through $x_i^\star$ (thus requiring a large number of fixed-point updates for the backward pass and resulting in poor quality gradients) and instead hope to explore that further in future work. Table 7 shows the time taken by JIIO v/s Adam on the DEQ model to perform optimization in the inner loop. Again, we observe that JIIO takes more than 2.7x lesser time to converge, demonstrating the main advantage of using JIIO for input optimization problems with DEQs.

| Algorithm/Model | 5-way, 1-shot |
|---|---|
| MAML | 98.7 |
| Reptile | 97.68 |
| iMAML, GD | 99.16 |
| JIIO-DEQ | 97.33 |

Table 6: Accuracy obtained on the 5-way, 1-shot task from the omniglot dataset

| Model | time taken (ms) |
|---|---|
| PGD : 20 iters | 28948 |
| JIIO : 100 iters | 11836 |

Table 7: Time taken by JIIO vs. Adam to perform inner-loop optimization in DEQ-based meta-learning tasks

## B.2 Adversarial training

We showed the adversarial training results for L2 perturbations with $\epsilon = 1$ in section 4.3. However, for CIFAR10, it is also common to use $\epsilon = 0.5$ with L2 perturbations. Thus, we also show the robust and clean accuracy of models trained with PGD and JIIO with $\epsilon = 0.5$ for CIFAR10 in Table 8. We again observe that the models trained with JIIO and PGD show similar robust and clean accuracies showing that JIIO is as effective as PGD towards finding adversarial examples while being about 3x faster.

| Datasets | Train (↓) Test (→) | Clean | PGD | JIIO |
|---|---|---|---|---|
| | Clean | $78.47 \pm 0.94$ | $4.54 \pm 1.33$ | $4.85 \pm 2.93$ |
| CIFAR | PGD | $68.48 \pm 0.81$ | $51.77 \pm 0.75$ | $50.14 \pm 0.74$ |
| | JIIO | $67.79 \pm 2.33$ | $51.39 \pm 0.56$ | $51.25 \pm 0.66$ |

Table 8: Comparison of adversarial training approaches on L2 norm perturbations with $\epsilon = 0.5$. The rows represent the training procedure and the columns represent the testing procedure

### B.3  Generative Models

We provide additional samples and reconstructions from the JIIO-MDEQ model trained on the CelebA 64x64 images in Figure 5. We also tried training the JIIO-MDEQ model with minor modifications (increasing latent dimensions to 384 and switching the downscaling factor to 4) from the original model and training it with 100 inner-loop JIIO iterations. The reconstructions from the resulting model can be seen in Figure 6. We observe that the reconstructions are blurry and speculate that it's likely due to the squared error loss used in training and the limited capacity of the latent space (which is constrained by the size of the post-hoc density model we wish to learn). Salimans et al. [63] have proposed more suitable losses and representation approaches for high dimensional images which we hope to test in future work to scale up our models to large scale generative modeling problems.

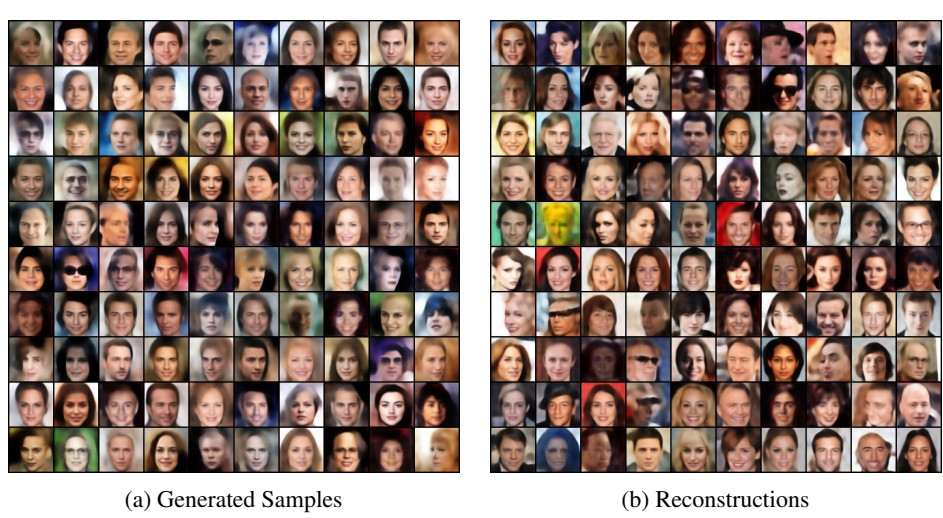

(a) Generated Samples                             (b) Reconstructions

Figure 5: Generated Samples and Reconstructions obtained from the JIIO-MDEQ generative model trained in Section 4 on CelebA 64x64 dataset.

### B.4  Inverse Problems

We showed results on 3 inverse problems in section 4. Figures 7, 8, 9, 10, 11, 12 show examples from unsupervised and supervised trained JIIO-MDEQ models on the tasks. We see that the unsupervised models approach the performance of the supervised alternatives on most tasks, however, the supervised approaches do tend to perform significantly better on the inpainting task.

## C  modeling and Training Details

In this section, we discuss various modeling decisions and training/testing details for all the tasks.

### C.1  Datasets

**Generative modeling and inverse problems.**  For all the tasks in the generative modeling and inverse problems sections, we work with the CelebA $64 \times 64$ dataset and use the standard train-

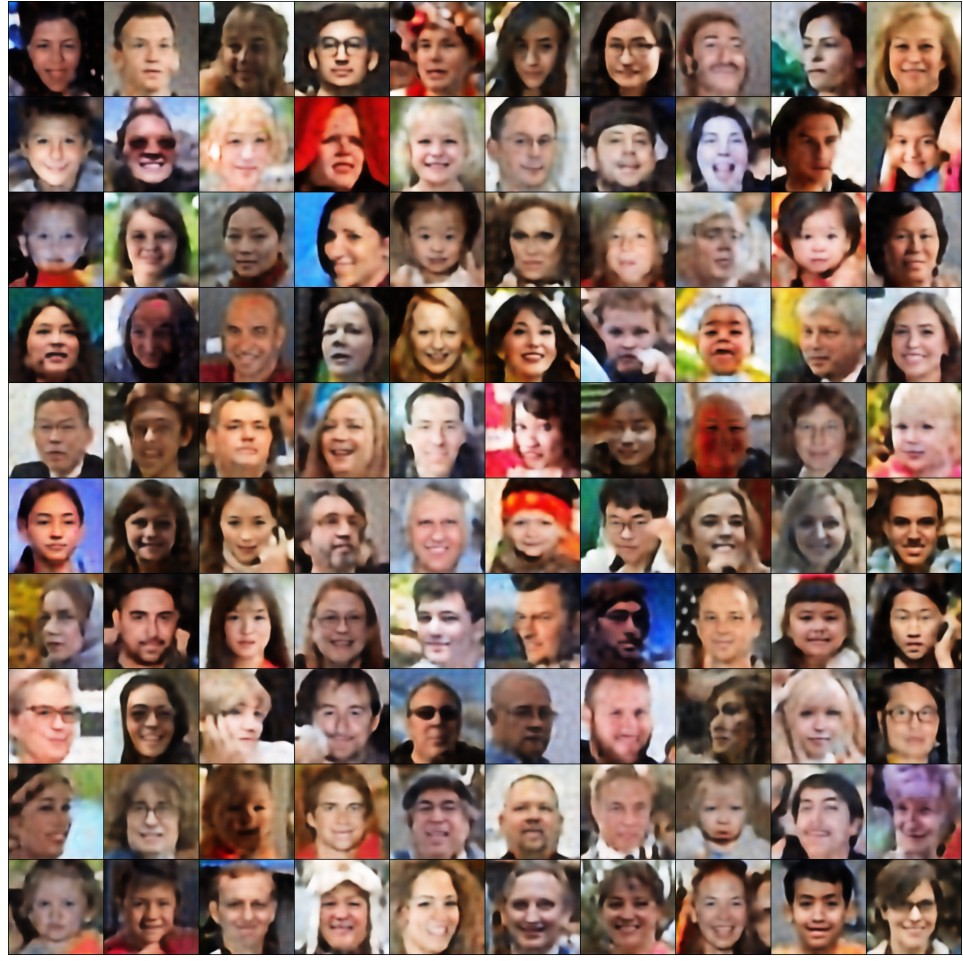

Figure 6: Reconstructions obtained from JIIO-MDEQ trained on 256x256 FFHQ dataset.

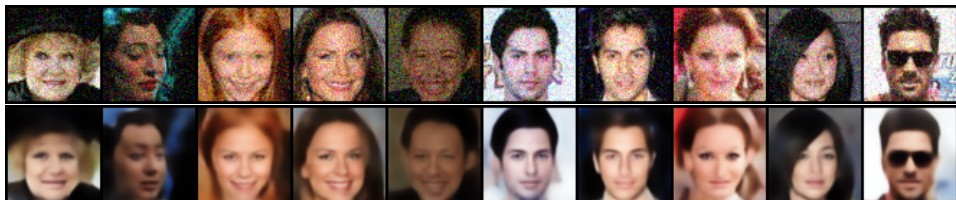

Figure 7: Supervised Image Denoising with additive noise sampled from $\mathcal{N}(0, 0.2)$: (top) Noisy image; (bottom) Recovered Image

val-test split prescribed in the original paper [51] with 162770 images in the training set, 19867 in the validation set and 19962 in the test set. We follow the procedure in [51] to crop and resize the images to obtain the $64 \times 64$ images from the dataset. In section B.3, we also show some preliminary reconstruction results on the FFHQ dataset with 70000 images aligned and cropped as done in [38] and then resized to $256 \times 256$. We additionally perform data augmentation on both datasets by performing random horizontal flips on each example.

**Adversarial training.** We use the well known CIFAR10 [48] and MNIST [50] datasets for our experiments on adversarial training. The CIFAR10 dataset consists of 60k $32 \times 32$ color images equally distributed over 10 classes, 50k of which are used for training and 10k for testing. The

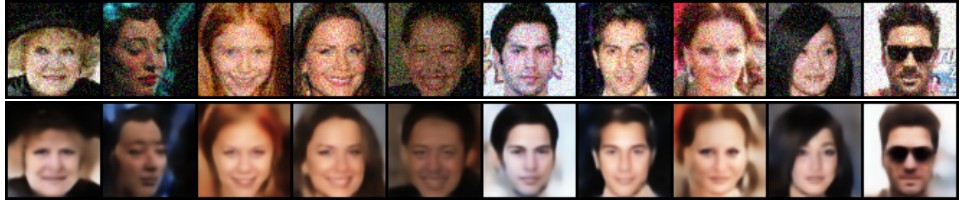

Figure 8: Unsupervised Image Denoising with additive noise sampled from $\mathcal{N}(0, 0.2)$: (top) Noisy image; (bottom) Recovered Image

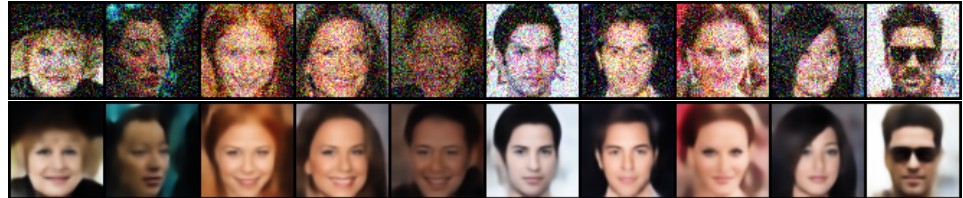

Figure 9: Supervised Image Denoising with additive noise sampled from $\mathcal{N}(0, 0.4)$: (top) Noisy image; (bottom) Recovered Image

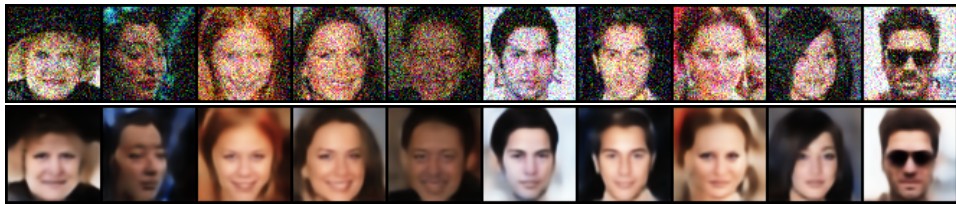

Figure 10: Unsupervised Image Denoising with additive noise sampled from $\mathcal{N}(0, 0.4)$: (top) Noisy image; (bottom) Recovered Image

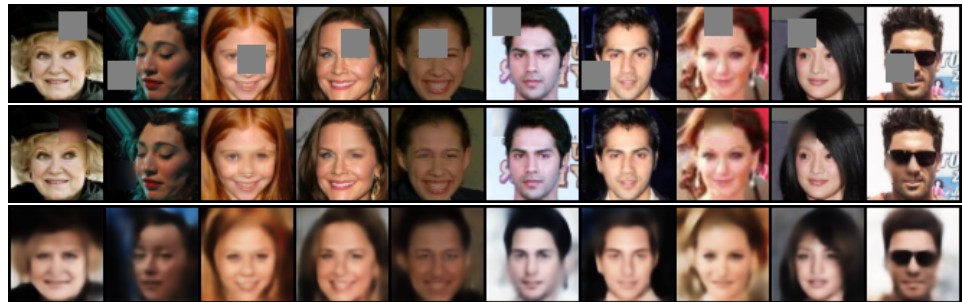

Figure 11: Supervised Image Inpaiting: (top) Incomplete image; (middle) Inpainted image; (bottom) Reconstructed Image

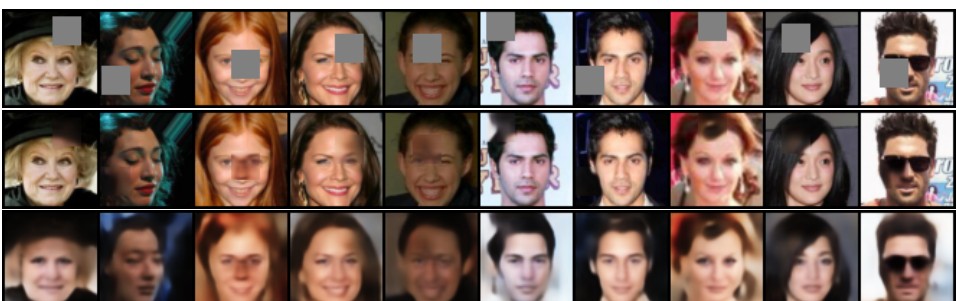

Figure 12: Unsupervised Image Inpaiting: (top) Incomplete image; (middle) Inpainted image; (bottom) Reconstructed Image

MNIST dataset consists of 70k grayscale $28 \times 28$ images equally distributed over 10 classes, with 60k used for training and 10k for testing.

**Gradient based meta-learning.** We use the Omniglot dataset [49] for our experiments with gradient based meta-learning. The Omniglot dataset contains 1623 different handwritten characters from 50 different alphabets, where each image is $28 \times 28$ dimensions. We create the tasks and the corresponding meta-training and meta-testing sets using the procedure described in [58].

## C.2 Architecture

For all experiments in the paper, we use the multiscale architectures proposed for DEQs (MDEQ) in Bai et al. [6]. In this section, we provide the specific instantiation and hyperparameters of the models used in each of our settings.

**Generative modeling and inverse problems.** We use a 4 branch MDEQ-LARGE model in all our experiments with a latent vector of size 128 dimensions mapped to the input injection for each branch using a single fully connected layer, while the output layer $h(z)$ merges the outputs of all the branches to obtain the output image as done in the Segmentation Networks in [6]. The hyperparameters of the network are provided in Table 9. Importantly, as mentioned in the experiments section of the paper, we replace all instances of Batch Normalization [35] with Group Normalization [70], in order to make the inner optimization independent for each example. Furthermore, for the MDEQ-VAE and MDEQ-AE baselines, we use a similar architecture for the encoder as well, except that the input injection and output layers are parameterized as in the input injection layers in the MDEQ classification networks in [6]. Additionally, for the baseline VAE, RAE and AE models, we borrow the architectures and code from [24].

For the generative modeling experiments, we additionally fit a density model on the inferred latent vectors inferred on the training set for sampling purposes. In our experiments, we fit a VAE with 4 fully connected layers each in the encoder and the decoder with the VAE latent dimension of size 384. The hidden dimensions for all non-latent layers of the VAE are set to 2096. We additionally also fit a 20 component GMM on the latents of the VAE given the larger latent dimension.

**Adversarial training.** We adopt a 2 branch MDEQ-SMALL model in all our experiments and use the structure of a standard classification network in [6]. The specific hyperparameters are provided in Table 9. As with experiments above, we replace all Batch Normalization layers with Group Normalization.

**Gradient based meta-learning.** We adopt a 2 branch MDEQ-SMALL model for our experiments and use the classification network proposed in [5]. As with experiments above, we replace all BatchNorm layers with GroupNorm. We additionally feed the task vector $x_i$ through a fully connected layer and use it as a film layer on top of the first GroupNorm in the residual block.

## C.3 Joint Inference and Input Optimization (JIIO)

As opposed to vanilla DEQ models, the joint optimization problem in the augmented DEQ requires a larger number of iterations and the corresponding acceleration techniques require larger memory sizes to converge than the forward inference in DEQ models. Moreover, the additional instability also benefits from additional damping. We accelerate the fixed point iterations using (Type-I) Anderson acceleration [4] in all our experiments. We run the optimization for the maximum number iterations as detailed in the experiments and pick the iterate with the least cost for each example in the batch.

**Generative modeling and inverse problems.** We use a memory size of 40 and observe that further increasing memory sizes can lead to faster convergence (at the cost of additional GPU memory requirements). Additionally, we damp the fixed point iterations with $\alpha = [\alpha_z, \alpha_\mu, \alpha_x] = [0.8, 0.6, 0.01]$. For experiments involving 100 inner loop iterations, we further reduce $\alpha_x = 0.003$ after 65 iterations in order to obtain finer solutions.

**Adversarial training.** We use a memory size of 20 for all tasks. For the experiments on CIFAR10 and MNIST, we use $\alpha = [\alpha_z, \alpha_\mu, \alpha_\delta] = [0.8, 0.6, 0.1]$ and $\alpha = [\alpha_z, \alpha_\mu, \alpha_\delta] = [0.8, 0.6, 0.6]$ respectively. Additionally, for MNIST, we reduce $\alpha_\delta = 0.2$ after 65 iterations. After each update, we additionally project the iterates $\delta$ onto an L2 ball with radius $\epsilon = 1$ in order to ensure the perturbations stay inside the L2 ball around the example.

**Gradient based meta-learning.** We use a memory size of 10 for this task and set $\alpha = [\alpha_z, \alpha_\mu, \alpha_x] = [0.8, 0.6, 0.04]$. As with the previous experiments we reduce $\alpha_x = 0.01$ after 65 iterations in order to obtain finer solutions. Additionally, we use a task/input vector of size 400 for the meta-learning task.

## C.4 Regularization

We use the regularization coefficient $\lambda = 0.1$ and 2 for the generative modeling and inverse problems experiments, and with 40 and 100 JIIO iterations, respectively. We use $\lambda = 0.01$ for the adversarial training experiments and $\lambda = 0.5$ for our meta-learning experiments. For all experiments, we compute the Hutchinson estimator $\mathbb{E}_{\epsilon \in \mathcal{N}(0,1)}[\epsilon^\top J_z^\top J_z \epsilon]$ using 2 samples of $\epsilon$.

## C.5 Compute and Runtime

The generative model and inverse problems experiments were trained on 4 RTX-2080 Ti GPUs. The generative modeling experiments were run for 50k training steps. For the inverse problems experiments, we trained models with 100 JIIO iterations for 25k training steps, taking 4.5-5 days for each training run. Adversarial training experiments with JIIO trained models take 9-10 hours for CIFAR10 on 3 RTX-2080 Ti GPUs while taking 5-6 hours on 2 RTX-2080 Ti GPUs for MNIST experiments. The models trained with projected gradient descent take 20-21 hours to train on 3 RTX-2080 Ti GPUs for the CIFAR10 experiments while taking 13-14 hours to train on 2 RTX-2080 gpus for the MNIST experiments. For our meta-learning experiments, the models trained using JIIO used 4 RTX-2080 Ti GPUs for roughly 2 days.

## C.6 Space complexity of the method

As pointed out earlier, JIIO has higher memory requirements than the corresponding ADAM version due to the higher memory sizes used in computing the fixed point. For example, in the generative modeling/inverse problems, optimization using JIIO requires 17.45 GB of GPU memory as opposed to vanilla Adam based optimization which simply costs 7.47 GB GPU memory for a batch of 48 images from celebA. However, for the adversarial training problems, the memory requirements were comparable - JIIO requires 2.19 GB memory as opposed to 2.15 for projected gradient descent for a batch with 96 images from CIFAR10.

| Hyperparameter | Adversarial Training | | Generative Model/Inverse Problems | |
|---|---|---|---|---|
| | MNIST | CIFAR10 | CelebA64 | Omniglot |
| Input Image Size | $28 \times 28$ | $32 \times 32$ | $64 \times 64$ | $28 \times 28$ |
| Batch Size | 96 | 96 | 48 | 80 |
| Optimizer | Adam | Adam | Adam | Adam |
| (Start) Learning Rate | 0.001 | 0.001 | 0.001 | |
| Nesterov Momentum | 0.9 | 0.9 | 0.9 | 0.9 |
| Weight Decay | 0 | 0 | 0 | 0 |
| Number of Scales | 2 | 2 | 4 | 2 |
| # of Channels for Each Scale | [24, 24] | [24, 24] | [32,64,128,256] | [64, 128] |
| Width Expansion (in the residual block) | 5$\times$ | 5$\times$ | 5$\times$ | 5$\times$ |
| Normalization (# of groups) | GroupNorm(8) | GroupNorm(8) | GroupNorm(8) | GroupNorm(8) |
| Weight Normalization | Yes | Yes | Yes | Yes |
| # of Downsamplings Before Equilirbium Solver | 0 | 0 | 0 | 0 |
| Forward Quasi-Newton Threshold $T_f$ | 18 | 18 | 18 | 18 |
| Backward Quasi-Newton Threshold $T_b$ | 20 | 20 | 20 | 20 |
| Broyden's Method Storage Size $m$ | 20 | 20 | 20 | 20 |
| Anderson JIIO Memeory Storage Size $M$ | 20 | 20 | 40 | 10 |
| Anderson JIIO Damping $\alpha$ | [0.8, 0.6, 0.1] | [0.8, 0.6, 0.6] | [0.8, 0.6, 0.01] | [0.8, 0.6, 0.04] |
| Variational Dropout Rate | 0.0 | 0.0 | 0.0 | 0.0 |

Table 9: MDEQ hyperparameters for each task