# OpenReview forum: "Joint inference and input optimization in equilibrium networks"
_NeurIPS.cc/2021/Conference — NeurIPS 2021 Poster_

### Official Review · Reviewer_XjMd · 2021-07-15

**Rating:** 6
**Confidence:** 4

**Summary:**

The paper proposes an extension to the DEQ framework that allows to simultaneously perform input and output optimization of the implicit DEQ layer. The extension called JIIO can itself be cast as another DEQ and backpropagated through. This appears to be useful in a number of applications such as decoder-only autoencoders where the encoder is defined implicitly as optimization over the latent representation.

**Limitations And Societal Impact:**

No issues.

**Main Review:**

Novelty:

The proposed method is novel to my knowledge.

Clarity:

The paper is well-written and easy to follow. I have some comments that are related to presentation of the method as well and I list them in the Quality section.

Quality:

I followed derivations of the optimization scheme and I find it sound.

My main point of concern is instability of the method which is important to properly investigate (if not fix completely) to make JIIO really useful to community.

1) I would like to have a better explanation for this rather than the trivial intuition that the optimization problem is more difficult. Especially given that some of the optimization methods simply failed to converge.
2) Similarly, it is unclear how to choose the output in JIIO given the optimization instability. Is the lowest-loss solution returned? Is it done both during training and during inference?
3) As an idea on the potential source of instability, did authors measure error in estimating the inverse Jacobian / its positive-definiteness?
4) Is it an artifact arising from using non-optimal step sizes?

Significance:

The list of considered applications is somewhat niche in the sense that DEQs could be not necessarily the best framework to target those, but I still think it's a valid contribution and JIIO will find more use in the community.

I'm keen to revise my rating based on authors reply.

**Time Spent Reviewing:**

5

---

> ### Author Response · Authors · 2021-08-10
> **Response to Reviewer XjMd**
>
> We thank the reviewer for the detailed comments and feedback. We try to address the main concerns of the reviewer as follows:
>
> 1) **Stability** - The stability issues were indeed a cause of concern with the JIIO optimization and training. We believe that there were multiple factors contributing to it and explored a couple of crucial ones:
>     * Choice of the optimizer/acceleration method - We show in section 7.1.1 in the appendix, that the choice of optimizer of acceleration method can be crucial towards obtaining stable convergence. For example, Anderson Type-II or the Augmented Lagrangian method (we performed those experiments but didn't include it in the plot) are much more stable compared to Anderson Type-I but required a much larger number of iterations to converge (about 2x to 3x). In fact, the relative instability of Anderson Type-I has also been discussed in previous work [3][4]. However, since speed was an important point of consideration for our experiments, we nevertheless went with Anderson Type-I and choose the output of the optimizer using a heuristic that trades off between a small kkt residual and a small cost. We observed that the criterion for this iterate selection can be somewhat flexible but preferably kept consistent between training and test runs when JIIO is used.
>     * High eigenvalues of the Jacobian of $f_\theta$ - It has been observed in previous work [1][2] that the stability of convergence of a DEQ is directly related to the spectral radius of the Jacobian. To that end, we regularize the jacobian trace of $f_\theta$ as pointed out in section 6.4 in the appendix. This has also been shown more recently in the context of DEQs more generally with much more extensive analysis in [2]. In the same spirit, we also tried to regularize the jacobian of the full ''augmented'' DEQ system. However, that did not help with improving stability much as the jacobian of $f_\theta$ didn't get sufficiently regularized as a result. However, we only conducted some preliminary experiments in that regard and it probably deserves further inspection.
>         Apart from explicitly regularizing the Jacobian, we also try to address the problem by adding additional damping as shown in Eq. (11) which helps mitigate the issues arising from large eigenvalues to some extent.
>
> 2) **Other applications** The reviewer also points out that the list of applications considered might not be the best suited for DEQs. We would just like to point out that since our submission to Neurips, we have also managed to test our approach on additional applications which inherently require solving a bi-level optimization problem like gradient based meta-learning and obtained decent performance. Also, as pointed out by Reviewer yNXE, we believe that JIIO would also be useful in the context of training EBMs.
>
> We hope the above answer addresses some of the concerns expressed by the reviewer.
>
>
> [1] Winston, E. and Kolter, J.Z., 2020. Monotone operator equilibrium networks. arXiv preprint arXiv:2006.08591
>
> [2] Bai, S., Koltun, V. and Kolter, J.Z., 2021. Stabilizing Equilibrium Models by Jacobian Regularization. arXiv preprint arXiv:2106.14342.
>
> [3] Zhang, J., O'Donoghue, B. and Boyd, S., 2020. Globally convergent type-I Anderson acceleration for nonsmooth fixed-point iterations. SIAM Journal on Optimization, 30(4), pp.3170-3197.
>
> [4] Fang, H.R. and Saad, Y., 2009. Two classes of multisecant methods for nonlinear acceleration. Numerical Linear Algebra with Applications, 16(3), pp.197-221.

---

### Official Review · Reviewer_EACo · 2021-07-17

**Rating:** 7
**Confidence:** 4

**Summary:**

The paper describes a method for optimizing the inputs of a deep equilibrium model (DEQ) at inference or training time by combining the input optimization procedure with the equilibrium finding procedure, transforrming the problem in an augmented DEQ.

The method is evaluated on several image-based tasks: generative model with optimized latents, inpainting/deblurring and generative adversarial training, yielding competitive results with baselines and a significant speedup compared to the typical multi-level optimization procedure.

**Limitations And Societal Impact:**

The authors discuss the main limitation which is the substantial memory penalty of their method.

**Main Review:**

The paper combines two active areas or research: implicit models, specifically DEQs, and forward-pass optimization on latent or input variables which is used on a variety of approaches.

It has been previously noted that  forward-pass optimization can be interpreted as computing the fixed point of an implicit model, here the authors show that when this is applied to a model which is already a DEQ in its base (unoptimized) forward pass, the computation of the DEQ internal fixed point and the external optimization fixed point can be combined as the computation of the fixed point of an augmented DEQ. This is more efficient than backpropagating through the DEQ and solving the optimization problem by gradient descent.

The research question is interesting, the paper is well written and easy to follow. The experimental results show comparable results with the baselines and substantial speedups. The tasks are reasonably varied. The only drawback is that no code release is planned.

Overall I think this is a good contribution to the field.

**Time Spent Reviewing:**

2

---

> ### Author Response · Authors · 2021-08-10
> **Response to Reviewer EACo**
>
> We're glad the reviewer liked our paper. We would just like to inform that we definitely plan to release the code publicly along with the paper if our paper gets accepted. Further, we note that the code is attached with the supplementary material in our submission as well.

---

### Official Review · Reviewer_yNXE · 2021-07-17

**Rating:** 7
**Confidence:** 4

**Summary:**

The paper argues that input optimization problems are pervasive in modern machine learning methods (adversarial example optimization, latent optimization in generative models, image completion/denoising). The authors then make a connection to deep equilibrium models (DEQ), which generate an output as a fixed-point of a single non-linear function. The connection is that both input optimization and DEQ process are fixed point iterations and can be solved jointly as an equality-constrained optimization problem. This results in 3-9x speeds on adversarial example generation and generative image modeling tasks.

**Limitations And Societal Impact:**

I discussed the limitation in Significance section of my main review.

**Main Review:**

Originality:
The idea of solving for input and equilibrium point optimization as a joint problem is to the best of my knowledge novel and is well-motivated - indeed input optimization with DEQs is wasteful if done naively as nested optimization and taking into account the structure of the problem allows for more efficient methods.

Quality:
The mathematical formulation is sound to the best of my knowledge.

Clarity:
The paper is clearly written and all technical components are well-motivated and given sufficient intuition. I especially appreciated an alternative interpretation of the formulation in section 3.3 as that more closely connected to my background and made it easier to reason about properties of this method.

The paper and appendix contain enough details to recreate the experiments.

As authors admit, the variable naming convention sometimes clashes with current generative modeling literature. While I understand the reasoning behind this, it would be nice to have an algorithm figure with all variables annotated so that the reader can quickly refer to for refresher on the meaning of variable names.

Significance:
DEQs are an intriguing class of models that deserve continuing investigation. I see the impact of this paper as two-fold: 1) it extends the capabilities of DEQ models by making input optimization tractable in these models 2) it gives an inspirational example of how DEQ models can be extended and combined with other mathematical tools.

The empirical results in generative image modeling on CelebA and FFHQ qualitatively don’t look very strong and quantitatively the VAE baseline is rather weak (HDCGAN reported FID of 8.44 on 64x64 CelebA generation in 2017, if I’m comparing the right benchmark). As such, I don’t see the community focused on best generative modeling performance adopting this work at the moment (of course, this is not the point of the paper, but it does limit significance).

The other two concerns I have about general adoption and impact of this work is stability of training and time and space complexity. For the latter, authors provide wallclock time comparisons and briefly mention memory footprint challenges. However, it would be very helpful to people considering building on this work to know time and space complexity of your algorithm, especially since memory could be a bottleneck in scaling this approach up.

For the concerns of stability, first I really appreciate authors for sharing their observations of stability issues rather than avoiding disclosing them - this is very helpful for readers to know and for presents opportunities for people to build on the work! Do the authors have any hypotheses for stability issues they could share in the paper? Is it potentially from fixed-point iteration not running to convergence? General instability of non-convex Lagrangian in eq 13? In this case, have you tried augmented Lagrangian known to improve stability? Any additional discussion here would be very helpful.

Towards applicability, I want to mention that another area where input optimization is prevalent recently is energy-based models. These models generate a datapoint as the minimum of the energy function (which could be a DEQ model) [A Tutorial on Energy-Based Learning by LeCun et al] and often use this optimization process in contrastive training of the models [i.e. Implicit Generation and Generalization in Energy-Based Models by Du et al, Divergence Triangle by Han et al, and many others].

But even on the first two merits alone, combined with the paper being sound I believe this work is worthy of sharing with our community and will inspire follow-up research.

**Time Spent Reviewing:**

3-4 hours

---

> ### Author Response · Authors · 2021-08-10
> **Response to Reviewer yNXE**
>
> We thank the reviewer for the detailed comments and feedback. We respond to the main concerns and suggestions brought up by the reviewer as follows:
>
> 1) **Notation and Figure** We acknowledge the reviewers concern regarding the notation in the case of generative models and would add an Algorithm figure for better clarity as suggested by the reviewer.
> 2) **Experiments** -  We address these concerns in Point (3) of our response to reviewer DfXs. - https://openreview.net/forum?id=RgH0gGH9B64&noteId=sl5APsDTFF1
> 3) **More details regarding stability** -  We address these concerns in Point (1) of our response to reviewer XjMd - https://openreview.net/forum?id=RgH0gGH9B64&noteId=1xash-NRUla
> 4) **Space complexity of the method** - We acknowledge the concerns regarding the space complexity of the method brought up by the reviewer. Our method indeed does require more memory resources than the corresponding baselines in the generative modelling/inverse problems - optimization using JIIO requires 17.45 GB of GPU memory as opposed to vanilla Adam based optimization which simply costs 7.47 GB GPU memory for a batch of 48 images from celebA. However, for the adversarial training problems, the memory requirements were comparable - JIIO requires 2.19 GB memory as opposed to 2.15 for projected gradient descent for a batch with 96 images from CIFAR10. We will add these discussions along with the discussions on time complexity of the method in section 4.1 and 4.1.
> 5) **Additional applications, EBMs** - We thank the reviewer for pointing out the applicability of the method towards EBM training. Indeed, that is one of the application areas we have been thinking of applying our approach in future work (specifically in the context of physics related applications - e.g in solving the schrodinger's equation for n-electron systems etc), however the main issue we foresee with such an application would be the need for adding additional noise at each optimization step in order to ensure proper mixing as acceleration methods like anderson acceleration might not work well in such settings. However, performing vanilla gradient descent with noise might still work well and would definitely be an interesting direction to explore. On a slightly tangential note, we have been experimenting with other applications as well like gradient based meta-learning where we have observed some promising initial results with JIIO used for the inner loop updates.
>
> We hope the above response addresses some of the concerns expressed by the reviewer.

---

### Official Review · Reviewer_DfXs · 2021-07-28

**Rating:** 5
**Confidence:** 4

**Summary:**

This submission considers the problem of joint inference and input optimization for DEQ model. The main contribution is that it can perform inference and input optimization simultaneously.


**Limitations And Societal Impact:**

No need to consider the Societal Impact

**Main Review:**

Pros:
The most important contribution of this submission is that it shows that there is no need to exactly finds the fixed point before performing the input optimization. In some special cases, the input p[timizaion won't introduce additional calculation since we can re-use the converged solution for the inner loop for parameters updating.

Cons:
1. Writing is too casual, and the motivation is very unclear. It is very unfriendly to readers who don't know much about DEQ and its related underlying problem.

2. The convergence guarantee for the problems, such as Eq.(5), relies on the assumptions on the  $\ell(\cdot)$ and $f_{\theta}$, which are all missing in this submission. Things should be more rigorous, especially for the mathematical process.

3. For experimental parts, there are too few methods to compare. The experiment mission is too simple, and the scale is too small, which cannot fully show the advantages of the proposed method.

**Time Spent Reviewing:**

4h

---

> ### Author Response · Authors · 2021-08-10
> **Response to Reviewer DfXs**
>
> We thank the reviewer for the detailed comments and feedback. We respond to the main criticisms of the work as follows:
>
> 1) **Writing and clarity** - We regret that the reviewer found the writing casual. We will try to improve the draft by making things more formal, providing a more comprehensive overview of DEQs and motivate the problem more concretely in the context of DEQs. We will also add an algorithm figure that provides a succinct overview of the method and provides notational clarity as suggested by Reviewer yNXE.
>
> 2) **Convergence guarantees and mathematical rigor** - In the paper, we chose not to address the convergence guarantees given the highly non-convex setting we are operating in, and the functions don't necessarily obey the required assumptions to guarantee convergence. We instead chose to just show that the solution to our optimization procedure (if converged) would correspond to the solution of the original optimization problem we care about (in Section 3.3).
>     However, in hindsight, we do agree with the reviewer that mentioning assumptions on $\ell()$ and $f_\theta$ might have been useful to provide intuitions on how to design them. In the final draft of the paper, we can add a discussion on  assumptions on $\ell()$ and $f_\theta$ required to ensure (local) convergence. Briefly, because the fixed point iterations we describe in our paper involve min-max optimization or bi-level optimization of non-convex-concave objectives, it is difficult to state a priori assumptions on the function that ensure local convergence.  However, just like gradient descent converges in locally convex regions, the simplest set of assumptions needed to ensure convergence in our setting is the requirement that that the _joint_ Jacobian of the fixed point iterations we describe be strongly monotone with smoothness parameter m and and Lipschitz constant L.  Then by standard arguments (see e.g., Section 5.1 of [1]), the fixed point iteration with step size $ \alpha < m/L^2$ will converge.  Note that these are substantially weaker rates and constants than required for typical gradient descent or the minimization of locally convex function because the coupling between the fixed point iterations introduce cross-terms in the Jacobian, and because the rates for fixed point iterations involving monotone operators are inherrently looser and require e.g. the strong monotonicity assumption which is not needed by minimization alone.  But we're certainly happy to add discussion of this point to the paper.
>
> 3) **Experiments** - (Also addressing concerns raised by Reviewer yNXE) In the experiments section, our main aim was to show that we can successfully train models using JIIO in various settings (at a much faster pace than performing vanilla bi-level optimization). Further, in the generative modelling setting, our aim was to introduce an entirely new class of generative model (powered by JIIO) and show that they obtain performance comparable to their auto-encoder based counterparts. However, we agree that our models don't outperform or match performance of current SOTA models (such as GANs, hierarchical VAEs or Score based models etc) in the generative modelling settings. We see our work as a first step in the direction and believe that with future improvements to our model (improvements to the stability and training speeds), we (and others) will be able to obtain much better performance and demonstrate significant improvements.
>
>
> We hope the above response addresses some of the concerns expressed by the reviewer.
>
> [1] https://web.stanford.edu/~boyd/papers/pdf/monotone_primer.pdf

---

### Decision · Program_Chairs · 2021-09-27

**Decision:**

Accept (Poster)

**Comment:**

The reviewers found the paper to be interesting, novel and well-written. This paper will likely be of interest to the community and inspire follow-on work. In review, the authors agreed to add a discussion on assumptions on assumptions on l() and f_theta required to ensure (local) convergence; we expect the authors to follow through on this.